# Protocol and Reagents for Pseudotyping Lentiviral Particles with SARS-CoV-2 Spike Protein for Neutralization Assays

**DOI:** 10.3390/v12050513

**Published:** 2020-05-06

**Authors:** Katharine H. D. Crawford, Rachel Eguia, Adam S. Dingens, Andrea N. Loes, Keara D. Malone, Caitlin R. Wolf, Helen Y. Chu, M. Alejandra Tortorici, David Veesler, Michael Murphy, Deleah Pettie, Neil P. King, Alejandro B. Balazs, Jesse D. Bloom

**Affiliations:** 1Division of Basic Sciences and Computational Biology Program, Fred Hutchinson Cancer Research Center, Seattle, WA 98109, USA; kdusenbu@fredhutch.org (K.H.D.C.); reguia@fredhutch.org (R.E.); adingens@fredhutch.org (A.S.D.); aloes@fredhutch.org (A.N.L.); kmalone2@fredhutch.org (K.D.M.); 2Department of Genome Sciences, University of Washington, Seattle, WA 98195, USA; 3Medical Scientist Training Program, University of Washington, Seattle, WA 98195, USA; 4Division of Allergy and Infectious Diseases, University of Washington, Seattle, WA 98195, USA; crwolf@uw.edu (C.R.W.); helenchu@uw.edu (H.Y.C.); 5Department of Biochemistry, University of Washington, Seattle, WA 98109, USA; tortorici@uw.edu (M.A.T.); dveesler@uw.edu (D.V.); neil@ipd.uw.edu (N.P.K.); 6Institute Pasteur & CNRS UMR 3569, Unité de Virologie Structurale, Paris 75015, France; 7Institute for Protein Design, University of Washington, Seattle, WA 98195, USA; murphymp@uw.edu (M.M.); ddpettie@gmail.com (D.P.); 8The Ragon Institute of Massachusetts General Hospital, the Massachusetts Institute Technology, and Harvard University, Cambridge, MA 02139, USA; abalazs@mgh.harvard.edu; 9Howard Hughes Medical Institute, Seattle, WA 98103, USA

**Keywords:** SARS-CoV-2, COVID-19, coronavirus, neutralization assay, lentiviral pseudotype, Spike, cytoplasmic tail, ACE2, 293T-ACE2, luciferase, ALAYT

## Abstract

SARS-CoV-2 enters cells using its Spike protein, which is also the main target of neutralizing antibodies. Therefore, assays to measure how antibodies and sera affect Spike-mediated viral infection are important for studying immunity. Because SARS-CoV-2 is a biosafety-level-3 virus, one way to simplify such assays is to pseudotype biosafety-level-2 viral particles with Spike. Such pseudotyping has now been described for single-cycle lentiviral, retroviral, and vesicular stomatitis virus (VSV) particles, but the reagents and protocols are not widely available. Here, we detailed how to effectively pseudotype lentiviral particles with SARS-CoV-2 Spike and infect 293T cells engineered to express the SARS-CoV-2 receptor, ACE2. We also made all the key experimental reagents available in the BEI Resources repository of ATCC and the NIH. Furthermore, we demonstrated how these pseudotyped lentiviral particles could be used to measure the neutralizing activity of human sera or plasma against SARS-CoV-2 in convenient luciferase-based assays, thereby providing a valuable complement to ELISA-based methods that measure antibody binding rather than neutralization.

## 1. Introduction

Infection with SARS-CoV-2 elicits antibodies that bind to the virus [1,2,3,4,5,6]. However, as is the case for all viruses [7,8,9,10], only some of these antibodies neutralize the virus’s ability to enter cells [4,5,11,12]. Whereas studies of immunity to SARS-CoV-2 are limited, for many other viruses, neutralizing antibodies are more strongly correlated with protection against reinfection or disease than antibodies that bind but do not neutralize [7,8,9,10,13,14,15]. Indeed, for other coronaviruses, neutralizing antibodies are protective in mouse models of infection [16,17,18,19,20] and associated with at least some reduced susceptibility to re-infection or disease in humans [15,21,22]. Furthermore, anecdotal reports have suggested that the passive transfer of neutralizing antibodies to sick patients may help alleviate disease from SARS-CoV-2 and its close relative, SARS-CoV [23,24,25].

However, while there are now well-characterized and high-throughput methods (such as ELISA assays) to measure total antibody binding to SARS-CoV-2 or some of its key constituent proteins [2,6,26], quantifying neutralizing antibody activity is more difficult. The most biologically relevant method is to directly measure how antibodies or sera inhibit infection of cells by replication-competent SARS-CoV-2. Such live-virus assays have now been performed to quantify neutralizing activity in the sera of infected patients or characterize the potency of individual antibodies [1,6,12,27]. However, the throughput and accessibility of live-virus neutralization assays with SARS-CoV-2 is limited by the fact that the virus is a biosafety-level-3 agent that must be worked with in specialized facilities.

An alternative approach that alleviates these biosafety limitations leverages the fact that all known neutralizing antibodies to SARS-CoV-2 (and other coronaviruses that lack a hemagglutinin-esterase protein) target the virus’s Spike protein [1,12,27]. Spike protrudes prominently from the surface of SARS-CoV-2 virions, and is necessary and sufficient to enable the virus to bind and enter cells [28]. Spike from several coronaviruses can be “pseudotyped” onto safer nonreplicative viral particles in place of their endogenous entry protein, thereby making entry of these particles into cells dependent on Spike [29,30,31,32,33,34,35,36]. For SARS-CoV-2, such pseudotyping has recently been reported using HIV-based lentiviral particles [4,27,37], MLV-based retroviral particles [12,38], and VSV [29,39,40,41]. In the data reported to date, results from such pseudovirus neutralization assays have correlated well with measurements made using live SARS-CoV-2 [1,12,27,39]. However, the detailed protocols and reagents to perform such assays are not yet widely available to the scientific community.

Here, we filled this gap by providing a detailed description of how to pseudotype lentiviral particles with Spike. We explained how these pseudotyped particles could be used to conveniently measure Spike-mediated cell entry via fluorescent or luciferase reporters, and to quantify the neutralizing activity of human plasma. Finally, we described all the necessary experimental reagents and make them available in the BEI Resources reagent repository (https://www.beiresources.org/).

## 2. Results

### 2.1. General Approach for Pseudotyping Lentiviral Particles with SARS-CoV-2 Spike

The basic strategy for pseudotyping HIV-1-derived lentiviral particles is shown in Figure 1A. It involves transfecting 293T cells with a lentiviral backbone plasmid encoding a fluorescent or luminescent reporter protein, a plasmid expressing Spike, and plasmids expressing the minimal set of lentiviral proteins necessary to assemble viral particles. The transfected cells then produce Spike-pseudotyped lentiviral particles that can be used to infect permissive cells that express the SARS-CoV-2 receptor protein, ACE2 [28,29,41,42].

We used an HIV-based lentiviral system to produce viral particles pseudotyped with Spike. As shown in Figure 1A, this system requires co-transfecting cells with a lentiviral backbone encoding the reporter protein(s), a plasmid expessing Spike, and plasmids encoding the other HIV proteins necessary for virion formation (Tat, Gag-Pol, and Rev). We used two different lentiviral backbones: One that used a CMV promoter to drive expression of just ZsGreen, and another that used a CMV promoter to drive expression of luciferase followed by an internal ribosome entry site (IRES) and ZsGreen (hereafter referred to as the ZsGreen and Luciferase-IRES-ZsGreen backbones).

The Spike protein was from SARS-CoV-2 strain Wuhan-Hu-1 using the NCBI-annotated start site [46], with the nucleotide sequence codon optimized for expression in human cells. We used three variants of Spike (Figure 1B). The first variant was simply the codon-optimized Spike. The second variant had two amino acid mutations to basic residues in Spike’s cytoplasmic tail (K1269A and H1271A) that changed the sequence of the five most C-terminal residues to ALAYT. This variant is hereafter referred to as Spike-ALAYT. The rationale for Spike-ALAYT was that for the original SARS-CoV, the two analagous mutations were shown to improve plasma membrane expression of Spike by eliminating an endoplasmic reticulum retention signal [47,48]. The third variant had the cytoplasmic tail of Spike replaced with that from influenza hemagglutinin (HA). This variant is hereafter referred to as Spike-HAtail. The rationale for Spike-HAtail was that for the original SARS-CoV, deleting Spike’s cytoplasmic tail or replacing it with that from other viruses was shown to improve pseudotyping efficiency [30,49,50,51]. We validated that there was expression of Spike on the surface of 293T cells transfected with plasmids expressing each of these three variants (Figure 1C).

The sequences of all of the Spike and lentiviral plasmids are in Appendix A, and the plasmids are available from BEI Resources (see Materials and Methods for BEI catalog numbers).

### 2.2. Target 293T Cells Constititutively Expressing Spike’s ACE2 Receptor

To create a target cell line that is efficiently infected by the SARS-CoV-2 Spike-pseudotyped lentiviral particles, we transduced 293T cells with a lentiviral vector expressing human ACE2 under an EF1a promoter (lentiviral backbone plasmid sequence is in Appendix A and is available from BEI Resources as item NR-52512). To create a clonal cell line from the bulk transduction, we sorted single transduced cells by flow cytometry and re-expanded into large populations (note that there was not a selectable marker in these cells). We identified an expanded clone that expressed high levels of ACE2 (Figure 2A). This ACE2 expression appears stable overtime and has not noticeably decreased through 12 passages at the time of writing. This clone is hereafter referred to as 293T-ACE2 and is available from BEI Resources as item NR-52511.

We validated that the 293T-ACE2 cells were susceptible to infection by SARS-CoV-2 Spike-pseudotyped lentiviral particles by incubating 293T-ACE2 and parental 293T with equivalent amounts of viral particles carrying ZsGreen. As shown in Figure 2B, all Spike-pseudotyped viruses could infect the 293T-ACE2 but not the 293T cells. Virus pseudotyped with VSV G, an amphotropic viral entry protein that is not dependent on ACE2, efficiently infected both cell lines (Figure 2B).

### 2.3. Titers of Pseudotyped Lentiviral Particles with Different Spike Cytoplasmic Tail Variants

To quantify the titers of lentiviral particles pseudotyped with each of the Spike variants, we produced particles with each of these Spikes, as well as a positive control using VSV G and a negative control without a viral entry protein. We first produced viral particles using the ZsGreen backbone, and titered by flow cytometry to determine the number of transducing particles per ml. As shown in Figure 3A, all Spike variants produced titers ≈10^4^ transduction units per mL. These titers were about two orders of magnitude lower than those achieved with VSV G, but we considered them to be encouragingly high given that lentiviral particles can be further concentrated by a variety of methods [52,53]. We then produced viral particles using the Luciferase-IRES-ZsGreen backbone and found that we could achieve titers of >10^6^ relative luciferase units (RLUs) per mL in 96-well plate infections (Figure 3B). This titer was again about two orders of magnitude lower than that achieved using VSV G. As expected, the magnitude of the fluorescent signal from ZsGreen was lower for the Luciferase-IRES-ZsGreen backbone than for the ZsGreen-only backbone (Figure 3C), since the ZsGreen in the former construct was driven by an IRES rather than the primary promoter.

### 2.4. Neutralization Assays with Spike-Pseudotyped Lentiviral Particles

We next used the Luciferase-IRES-ZsGreen viruses to perform neutralization assays in 96-well plates. Because <10^5^ RLUs per well of a 96-well plate are necessary to achieve a signal >1000-fold above the background luciferase activity of virus-only controls, this assay required only a relatively modest volume of virus for a full 96-well plate neutralization assay.

We performed neutralization assays using plasma from a confirmed SARS-CoV-2-infected patient collected at 19 days post-symptom onset, and with soluble ACE2 protein fused to an IgG Fc domain (which neutralizes SARS-CoV-2 by acting as a decoy receptor [54]). As negative controls (not expected to neutralize), we used sera collected prior to the emergence of SARS-CoV-2 in late 2019. For these assays, we first made serial dilutions of the plasma, sera, or soluble ACE2-Fc in a 96-well plate. We then incubated these dilutions for 60 min with a volume of pseudotyped lentiviral particles sufficient to achieve 2 × 10^5^ RLUs of luciferase signal per well. Finally, we added the mix to a pre-seeded plate of 293T-ACE2 cells. We measured the luciferase signal at 60 h post-infection (see Materials and Methods for a more detailed protocol).

Both the plasma from the confirmed SARS-CoV-2-infected patient and the soluble ACE2-Fc effectively neutralized the virus (Figure 4A,B). For the plasma, the inhibitory concentrations 50% (IC50s) were ≈1:1600 (±25%) for all three Spike variants, which is in the range of values that have been reported for sera and plasma from other SARS-CoV-2 patients at a similar time post-infection [4]. For soluble ACE2-Fc, the IC50 was ≈2 μg/mL with the Spike and Spike-ALAYT-pseudotyped lentiviral particles, but was notably lower for the Spike-HAtail pseudotyped lentiviral particles (Figure 4B). Our measurement of ≈ 2 μg/mL for the unmodified Spike is higher than a previously reported IC50 of 0.1 μg/mL for soluble ACE2-IgG [54]. We suspect that the difference could be because our 293T-ACE2 target cells expressed high levels of ACE2, making them more resistant to neutralization by soluble ACE2. As expected, there was no neutralization of the pseudotyped virus by either pooled or individual human sera collected at dates prior to the emergence of SARS-CoV-2 (Figure 4C).

Our results are equivocal as to whether the cytoplasmic tail modifications greatly alter neutralization sensitivity. For the plasma neutralization, all three Spike variants (Spike, Spike-ALAYT, and Spike-HAtail) exhibited similar neutralization profiles (Figure 4A). However, for the soluble ACE2, the Spike-HAtail virus was notably more neutralization sensitive than the other two Spike variants (Figure 4B). Whereas the mechanism underlying the distinct neutralization sensitivity observed is unclear, it is possible that modifying the Spike’s cytoplasmic tail may alter opening of the receptor-binding domains [28]. Therefore, we suggest performing the assays using the Spike without any cytoplasmic tail modifications, particularly since none of the modifications tested here greatly improved pseudotyped lentiviral particle titers.

## 3. Discussion

We described a detailed protocol for producing SARS-CoV-2 Spike-pseudotyped lentiviral particles and performing neutralization assays. Although this basic pseudotyping approach has been described previously [4,12,27,29,37,38,39,40,41], we provided the first detailed protocol that made all reagents available in a public repository (https://www.beiresources.org/). We hope this protocol and reagents will more easily enable others to assess the neutralizing activity of antibodies and sera reactive to SARS-CoV-2.

We also found that modifying the cytoplasmic tail of SARS-CoV-2 Spike did not greatly improve titers of Spike-pseudotyped lentiviral particles. Indeed, one cytoplasmic tail modification we tested potentially altered the neutralization sensitivity of the pseudotyped lentiviral particles, suggesting that it may be undesirable. Whereas we did not test the full suite of cytoplasmic tail modifications that have been used for pseudotyping with Spike from the original SARS-CoV [30,49,50,51], our results suggest that modifications to the cytoplasmic tail of the SARS-CoV-2 Spike should be tested with caution.

Overall, we have described an easily accessible assay to study neutralizing antibody responses to SARS-CoV-2 in a biosafety-level-2 laboratory. This assay allows human sera or plasma samples to be screened in a convenient 96-well format, which will help facilitate the testing of large numbers of patient samples to better understand the development of immunity and to potentially screen donors for passive transfer of convalescent plasma [25,55].

## 4. Materials and Methods

### 4.1. Plasmids

The sequences of all plasmids used in this study are available in Genbank format in Appendix A and are also at https://github.com/jbloomlab/SARS-CoV-2_lentiviral_pseudotype/tree/master/plasmid_maps. The plasmids themselves are available in BEI Resources (https://www.beiresources.org/) with the following catalog numbers:pHAGE2-EF1aInt-ACE2-WT (BEI catalog number NR52512): Lentiviral backbone plasmid expressing the human ACE2 gene (GenBank ID for human ACE2 is NM_021804) under an EF1a promoter with an intron to increase expression.HDM-IDTSpike-fixK-HA-tail (BEI catalog number NR52513): Plasmid expressing under a CMV promoter the Spike from SARS-CoV-2 strain Wuhan-Hu-1 (Genbank NC_045512) codon-optimized using IDT, with the Spike cytoplasmic tail replaced by that from the HA protein of A/WSN/1933 (H1N1) influenza, and the Kozak sequence in the plasmid fixed compared to an earlier version of this plasmid.HDM-IDTSpike-fixK (BEI catalog number NR-52514): Plasmid expressing under a CMV promoter the Spike from SARS-CoV-2 strain Wuhan-Hu-1 (Genbank NC_045512) codon-optimized using IDT and the Kozak sequence in the plasmid fixed compared to an earlier version of this plasmid.HDM-nCoV-Spike-IDTopt-ALAYT (BEI catalog number NR-52515): plasmid expressing under a CMV promoter the Spike from SARS-CoV-2 strain Wuhan-Hu-1 (Genbank NC_045512) codon-optimized using IDT, with the Spike containing two mutations in the cytoplasmic tail such that the last five amino acids are ALAYT.pHAGE-CMV-Luc2-IRES-ZsGreen-W (BEI catalog number NR-52516): Lentiviral backbone plasmid that uses a CMV promoter to express luciferase followed by an IRES and ZsGreen.HDM-Hgpm2 (BEI catalog number NR-52517): lentiviral helper plasmid expressing HIV Gag-Pol under a CMV promoter.HDM-tat1b (NR-52518): Lentiviral helper plasmid expressing HIV Tat under a CMV promoter.pRC-CMV-Rev1b (NR-52519): Lentiviral helper plasmid expressing HIV Rev under a CMV promoter.pHAGE2-CMV-ZsGreen-W (NR-52520): Lentiviral backbone plasmid that uses a CMV promoter to express ZsGreen.

Note that all of these plasmids have ampicillin resistance. The only plasmid used in this study that was not in the BEI Resources catalog is the HDM-VSVG plasmid that expressed VSV G under a CMV promoter, and was used to create the positive control lentivirus pseudotyped with VSV G. However, numerous VSV G expressing plasmids are available from AddGene and other repositories.

### 4.2. Creation of 293T ACE2 Cells

VSV G-pseudotyped lentivirus packaging the human ACE2 was generated via co-transfecting 293T cells (ATCC, CRL-3216) with the pHAGE2-EF1aInt-ACE2-WT plasmid (Appendix A) and lentiviral helper plasmids (HDM-VSVG, HDM-Hgpm2, HDM-tat1b, and pRC-CMV-Rev1b). The resulting lentivirus was used to infect more 293T cells in the presence of 5 μg/mL polybrene. The transduced cells were stained with anti-human ACE-2 polyclonal goat IgG (AF933, R&D Systems, Minneapolis, MN, USA) primary antibody at 1 μg/mL and donkey anti-goat IgG conjugated to Alexa Fluor 488 (ab150129, Abcam, Cambridge, UK) secondary antibody at a 1:2500 dilution and sorted based on antibody staining. Once single cell clones had grown sufficiently, they were screened for ACE2 expression via flow cytometry and a clone with high expression was expanded and named 293T-ACE2 (Figure 2A). For verifying expression via flow cytometry, cells were harvested with enzyme-free dissociation buffer (13151014, ThermoFisher, Waltham, MA, USA) and stained with the anti-human ACE-2 polyclonal goat IgG primary antibody at 2 μg/mL and donkey anti-goat IgG (Alexa Fluor 488) secondary antibody at a 1:1000 dilution. For each staining step, cells were incubated with antibody in the dark at 4 °C for 30 min. Cells were washed three times with 3% BSA in PBS following each stain.

The 293T-ACE2 cells can be grown in D10 growth media (DMEM with 10% heat-inactivated FBS, 2 mM l-glutamine, 100 U/mL penicillin, and 100 μg/mL streptomycin) at 37 °C and 5% carbon dioxide. Note that there was not a selectable marker for the ACE2 expression. We found that ACE2 expression remained stable overtime (Figure 2A), but if there is a concern about expression, ACE2 levels can be periodically re-confirmed via antibody staining and flow cytometry. The 293T-ACE2 cells are available from BEI Resources as catalog number NR-52511.

### 4.3. Detailed Protocol for Generation of Pseudotyped Lentiviral Particles

Pseudotyped lentiviruses can be generated by transfecting 293Ts as depicted in Figure 1A. We used the following protocol:Seed 293T cells in D10 growth media (see Section 4.2 for media composition) so that they will be 50%–70% confluent the next day. For a six-well plate, this is 5 × 10^5^ cells per well (2.5 × 10^5^ cells per mL).At 16-24 h after seeding, transfect the cells with the plasmids required for lentiviral production. We transfected using BioT (Bioland Scientific, Paramount, CA, USA) following the manufacturer’s instructions and using the following plasmid mix per well of a six-well plate (plasmid amounts should be adjusted for larger plates):
1 μg of lentiviral backbone–we used either the ZsGreen (NR-52520) or the Luciferase-IRES-ZsGreen (NR-52516) backbone0.22 μg each of plasmids HDM-Hgpm2 (NR-52517), pRC-CMV-Rev1b (NR-52519), and HDM-tat1b (NR-52518)0.34 μg viral entry protein—either SARS-CoV-2 Spike (NR-52513, NR-52514, or NR-52515), VSV G (positive control), or transfection carrier DNA (E4881, Promega, Madison, WI, USA) as a negative control.At 18 to 24 h post-transfection, change the media to fresh, pre-warmed D10.At 60 h post transfection, collect virus by harvesting the supernatant from each well and filtering it through a 0.45 μm SFCA low protein-binding filter. Virus can be stored at 4 °C for immediate use or frozen at −80 °C. The titers of Spike- and VSV G-pseudotyped lentiviruses were found to be unaffected by a single freeze-thaw cycle (data not shown). Nonetheless, we recommend freezing virus in small aliquots to avoid multiple freeze-thaw cycles. All titers presented here are from virus that was frozen at −80 °C prior to use and underwent a single freeze-thaw.

### 4.4. Detailed Protocol for Titering Pseudotyped Lentiviral Particles

To determine viral titers, we used either flow cytometry (for viruses packaging the ZsGreen backbone) or a luciferase assay (for viruses packaging the Luciferase-IRES-ZsGreen backbone). A detailed titering protocol is described below and differences between these two readouts are noted:
Coat a 96-well cell-culture plate with 25 μL poly-l-lysine per well (P4707, Millipore Sigma, Burlington, MA, USA) according to the manufacturer’s protocol. Poly-L-lysine improves cell adherence and prevents cell disruption during infection.Seed a poly-l-lysine-coated 96-well plate with 1.25 × 10^4^ 293T-ACE2 cells per well in D10 media.The next day (12–24 h post-seeding), count at least two wells of cells to determine the number of cells present at infection.Prepare serial dilutions of the viruses to be titered in a final volume of 150 μL D10 growth media.
For ZsGreen virus, we started with a 1:5 dilution and made three 1:5 serial dilutions.For Spike-pseudotyped Luciferase_IRES_ZsGreen virus, we started with undiluted virus and made three 1:3 dilutions. For VSV G-pseudotyped Luciferase_IRES_ZsGreen virus, we started with a 1:50 dilution.Gently remove the media from the cells and slowly add the virus dilutions.Add polybrene (TR-1003-G, Sigma Aldrich, St. Louis, MO, USA) to a final concentration of 5 μg/mL. We did this by adding 3 μL of polybrene diluted to 250 μg/mL to our final infection volume of 150 μL. Polybrene is a polycation that helps facilitate lentiviral infection through minimizing charge-repulsion between the virus and cells [56].At 48–60 h post-infection, collect cells for analysis:
For flow cytometry:
Look at the cells under a fluorescent microscope and select wells that appear ~1%–10% positive. Harvest cells from these wells using trypsin and transfer them to a V-bottom plate or microcentrifuge tubes. Pellet cells at 300× *g* for 4 min and wash twice with 3% BSA in PBS. After the final wash, resuspend in 1% BSA in PBS and analyze via flow cytometry. We used a Becton Dickinson Celesta cell analysis machine with a 530/30 filter to detect ZsGreen in the FITC channel. Resulting FCS files were analyzed using FlowJo (v10) (BD Life Sciences, Ashland, OR, USA).Calculate titers using the Poisson formula. If P is the percentage of cells that are ZsGreen positive, then the titer per ml is: *-ln(1 − P/100) × (number of cells/well)/(volume of virus per well in mL)*. Note that when the percentage of cells that are ZsGreen positive is low, this formula is approximately equal to: *(% ZsGreen positive/100) × (number of cells per well)/volume of virus per well in mL.* Furthermore, the titers using even the Poisson equation will only be accurate if the percentage of cells that are ZsGreen positive is relatively low (ideally 1–10%).For luciferase:
Thaw luciferase reagent at room temperature. We used the Bright-Glo Luciferase Assay System (E2610, Promega, Madison, WI, USA).Prepare cells by removing 100 μL media from each well. Accounting for evaporation, this leaves ~30 μL of media in each well.Add 30 μL of luciferase reagent, mix well, and transfer all 60 μL to a black-bottom plate (Costar 96-well solid black, 07-200-590, Fisher Scientific, Waltham, MA, USA).Incubate plate for 2 min at room temperature in the dark, then measure luminescence using a plate reader. We used a Tecan Infinite M1000 Pro plate reader with no attenuation and a luminescence integration time of 1 s.Plot RLUs vs. virus dilution. Select an amount of virus for further assays where there is sufficient (>1000-fold) signal above virus-only background and a linear relationship between virus added and RLU.

### 4.5. Detailed Protocol for Neutralization Assays

The following protocol was developed to streamline neutralization assays with Spike-pseudotyped lentiviruses. This protocol can be performed with either human sera or plasma, or monoclonal antibodies. Note that for safety, sera or plasma should be heat-inactivated in a biosafety cabinet prior to use as described in Section 4.6.
Seed a poly-L-lysine-coated 96-well plate with 1.25 × 10^4^ 293T-ACE2 cells (BEI NR-52511) per well in 50 μL D10 (2.5 × 10^5^ cells per mL). Plan to infect this plate 8–12 h post-seeding.About 1.5 h prior to infecting cells, begin preparing serum and/or ACE2 dilutions in D10:
In a separate 96-well “setup” plate, serially dilute serum samples, leaving 60 μL diluted serum in each well. For the data in Figure 4A,C, we started at an initial serum dilution of 1:80 and did serial 2.5-fold dilutions, meaning each replicate of the assay required 5 μL of sera. For ACE2 (Figure 4B), we started with a concentration of 200 μg/mL and did serial three-fold dilutions.Add 60 μL of D10 to wells corresponding to virus only and virus plus cells control wells. Add 120 μL of D10 to media only and cells only control wells. See Figure 5 for an example plate layout.Dilute virus to ~2–4 × 10^6^ RLU per mL. Add 60 μL of diluted virus to all wells containing serum dilutions and the virus only and virus plus cells control wells.Incubate virus and serum at 37 °C for 1 h.Carefully add 100 μL from each well of the setup plate containing the sera and virus dilutions to the corresponding wells of the plate of 293T-ACE2 cells.Add polybrene (Sigma Aldrich, TR-1003-G) as described in Section 4.4 for a final concentration of 5 μg/mL in each well.Incubate at 37 °C for 48–60 h before reading out luminescence or fluorescence as described in Section 4.4.Plot the data. For our analysis, we first subtracted out the background signal (average of the “virus only” and “virus + 293Ts” wells) and then calculated the “maximum infectivity” for each plate as the average signal from the wells without serum (“virus + cells” wells). We then calculated the “fraction infectivity” for each well, as the luciferase reading from each well divided by the “maximum infectivity” for that plate. For the curves shown in Figure 4, we then fit and plotted the fraction infectivity data using the neutcurve Python package (https://jbloomlab.github.io/neutcurve/). This package fits a three-parameter Hill curve, with the top baseline being a free parameter and bottom baseline fixed to zero.

### 4.6. Human Plasma Sample and Soluble ACE2

The human plasma sample used in Figure 4A was collected at 19 days post-symptom onset from a patient with a confirmed SARS-CoV-2 infection. Prior to use, the plasma was heat-inactivated in a biosafety cabinet at 56 °C for one hour. This duration of heat treatment has been shown to be sufficient to inactivate SARS-CoV-2 [26,57], which has also not been reported to be present at high titers in the blood [58,59]. The negative control serum pools came from Gemini Biosciences, West Sacramento, CA, USA (Cat:100-110). The naïve serum pool collected in 2017–2018 is lot H86W03J. The age-matched negative control serum comes from serum residuals collected by Bloodworks Northwest. It was collected on 12/19/1989 and stored at −80 °C.

Soluble human ACE2 protein fused to the Fc region of human IgG1 was produced as described by the authors of [28]. This ACE2-Fc fusion protein was used in Figure 4B.

## Figures and Tables

**Figure 1 viruses-12-00513-f001:**
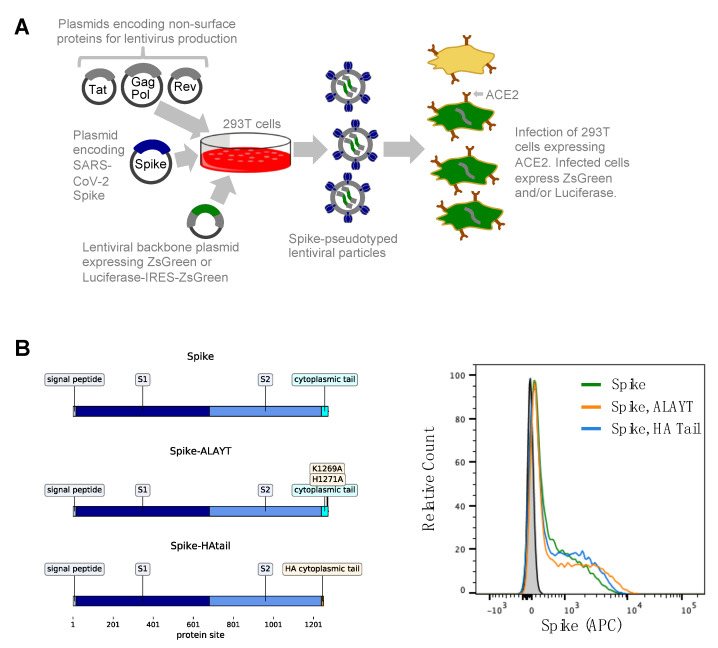
General approach for lentiviral pseudotyping. (**A**) 293T cells are transfected with a plasmid encoding a lentiviral backbone (genome) expressing a marker protein, a plasmid expressing Spike, and plasmids expressing the other HIV proteins needed for virion formation (Tat, Gag-Pol, and Rev). The transfected cells produce lentiviral particles with Spike on their surface. These viral particles can infect cells that express the ACE2 receptor. (**B**) We used three variants of Spike: The codon-optimized Spike from SARS-CoV-2 strain Wuhan-Hu-1, a variant containing mutations K1269A and H1271A in the cytoplasmic tail (such that the C-terminal five amino acids are ALAYT), and a variant in which the cytoplasmic tail of Spike has been replaced with that from influenza hemagglutinin (HA). (**C**) Spike expression on the surface of 293T cells transfected with the plasmids expressing our three Spike constructs was measured using flow cytometry 24 h post-transfection. Spike expression was measured by staining with in-house produced CR3022 antibody [43,44,45] at a concentration of 10 μg/mL followed by staining with an anti-human Fc antibody conjugated to APC (Jackson Labs, 109-135-098) at a 1:100 dilution.

**Figure 2 viruses-12-00513-f002:**
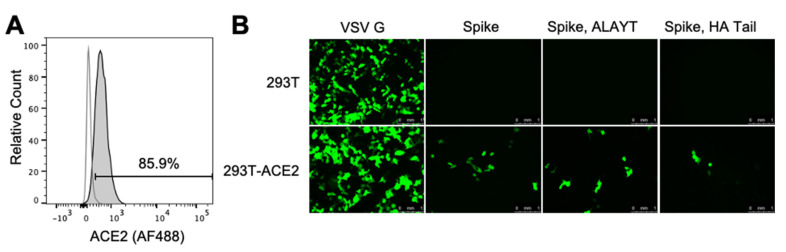
293T-ACE2 cells are infectable with SARS-CoV-2 Spike-pseudotyped lentiviral particles. (**A**) Flow cytometry plot showing expression of human ACE2 by the 293T-ACE2 cells (grey shaded) at passage 12 compared to parental 293T cells (white fill) as quantified by staining with an anti-ACE2 antibody (see Section 4.2 for detailed methods). The gate was set so the parental 293T cells were 2% positive. (**B**) Microscope images showing ZsGreen expression in 293T-ACE2 or 293T cells at 58 h after incubation with Spike- or VSV G-pseudotyped lentiviral particles with the ZsGreen backbone. For each viral entry protein, 293T and 293T-ACE2 cells were incubated with equal volumes of virus. Cells were incubated with 1/20^th^ the volume of VSV G-pseudotyped lentivirus compared to Spike-pseudotyped lentivirus. The decrease in infected cells for the Spike-HAtail virus compared to the other Spike-pseudotyped lentiviruses is consistent with this virus having somewhat lower titers (see Figure 3A).

**Figure 3 viruses-12-00513-f003:**
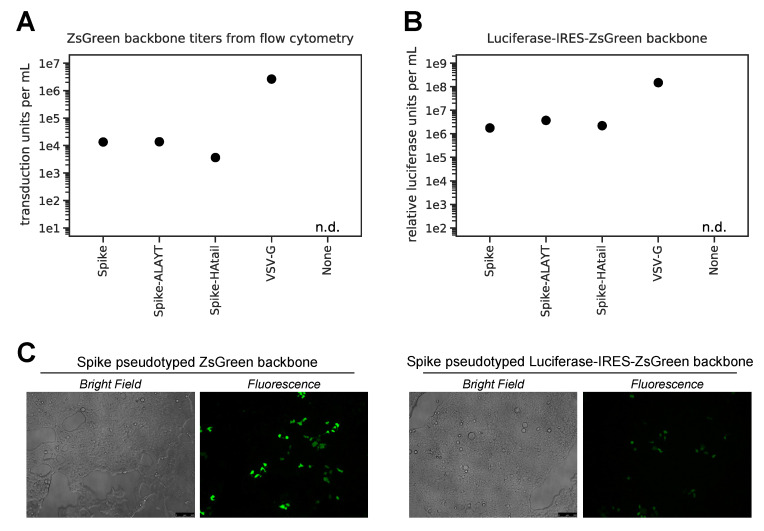
Titers of Spike-pseudotyped lentiviral particles in 293T-ACE2 cells. (**A**) Titers of the ZsGreen backbone pseudotyped with the three Spike variants or VSV G, as determined by counting green cells via flow cytometry analysis at 48 h post-infection and then calculating transduction-competent viral particles per mL from the percentage of green cells. The “n.d.” indicates that the titer was not detectable. Data shown are from a single representative example. (**B**) Titers of the Luciferase-IRES-ZsGreen backbone as determined by measuring relative luciferase units (RLUs). RLUs were determined at 48 h after infecting ~2.3 × 10^4^ 293T-ACE2 cells per well in 96-well plates. The RLUs per mL for the Spike-pseudotyped viruses are the average of three three-fold serial dilutions of virus starting at 50 μL virus in a total volume of 150 μL. For the VSV G-pseudotyped virus, RLUs per mL were averaged from two three-fold dilutions starting at 3 μL virus in a total volume of 150 μL. (**C**) Microscope images showing 293T-ACE2 cells transduced with Spike pseudotyped virus with either the ZsGreen or Luciferase-IRES-ZsGreen backbone at 60 h post-infection. As can be seen from the images, the ZsGreen backbone gave a stronger fluorescent signal than the Luciferase-IRES-ZsGreen backbone, presumably because this protein was expressed more strongly as the sole CMV-promoter driven transcript than as the second transcript driven by an IRES.

**Figure 4 viruses-12-00513-f004:**
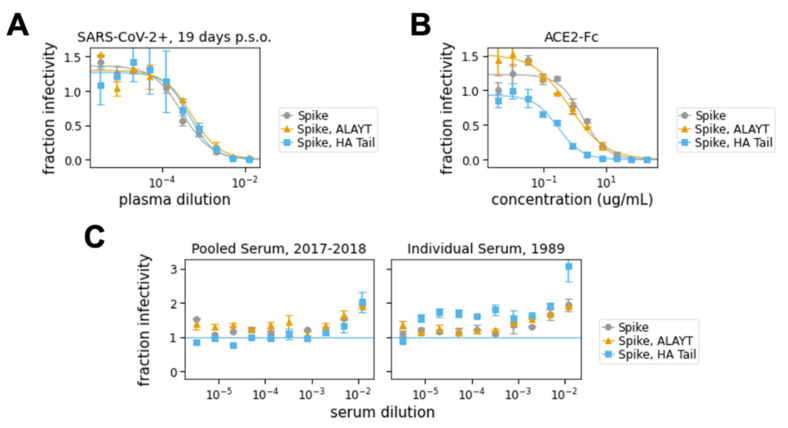
Neutralization assays. (**A**) Neutralization assay using plasma collected from a confirmed SARS-CoV-2-infected patient at 19 days post-symptom onset (“p.s.o.”). The IC50 for this plasma was 1:2076 for the Spike-pseudotype, 1:1334 for the Spike-ALAYT-pseudotype, and 1:1605 for the Spike, HAtail-pseudotype. (**B**) Neutralization assay using soluble ACE2 protein fused to the Fc domain from IgG (ACE2-Fc). The IC50 for ACE2-Fc was 2.49 μg/mL for the Spike-psdeudotype, 1.75 μg/mL for the Spike-ALAYT-pseudotype, and 0.25 μg/mL for the Spike-HAtail pseudotype. (**C**) Negative control sera collected prior to the emergence of SARS-CoV-2 did not neutralize the Spike-pseudotyped lentiviral particles. The serum from 1989 was from a person of a similar age at the time of serum collection as the confirmed SARS-CoV-2 infected patient whose plasma was tested in **A**. High concentrations of naïve serum seemed to enhance luciferase signal, perhaps because of components that improve cell-growth. Each point shows the average of duplicate values with error bars showing standard error.

**Figure 5 viruses-12-00513-f005:**
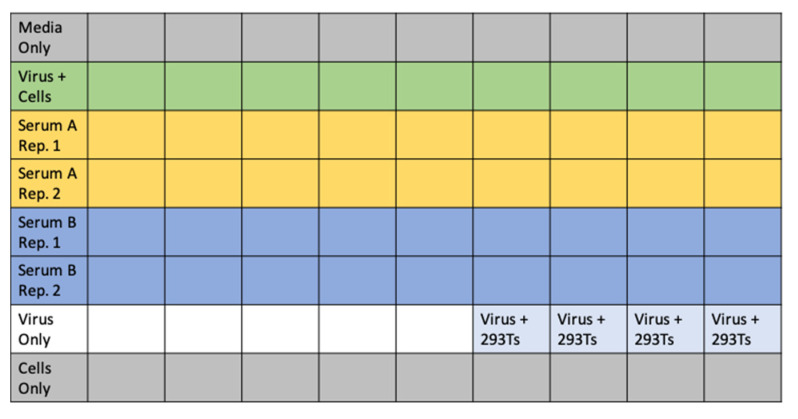
Example plate layout for neutralization assays. It is possible to run full-dilution series of two sera or plasma samples in duplicate on each plate with the necessary controls. These controls include media only, cells only, and virus-only wells, as well as four wells of virus-infecting 293T cells to confirm the lack of infection with Spike-pseudotyped lentivirus in the absence of ACE2. The average signal from the “Virus Only” and “Virus + 293Ts” wells provides the background signal. The “Virus + Cells” wells represent maximum infection without any serum and provide a metric for 100% virus infectivity. Note that “cells” here refers to the 293T-ACE2 cells. The different colors simply denote different conditions, such as different serum samples or cells. These conditions are labeled in the left-most column of the figure (or in each individual cell for the “Virus + 293Ts” condition).

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
