# Peer review of "Protocol and Reagents for Pseudotyping Lentiviral Particles with SARS-CoV-2 Spike Protein for Neutralization Assays"

_viruses, 2020, doi:10.3390/v12050513_

Round 1
Reviewer 1 Report
The Crawford et al manuscript is a description of a system for generating lentiviral particles pseudotyped by the SARS-Cov2 spike protein for use in neutralizing antibody assays. This is an important and timely article and will be very useful in the field. In addition to providing a very useful and detailed protocol along with easily accessible reagents, it shows that alterations of the tail of the spike protein (which has been previously done to increase titers of other coronavirus-pseudotyped virus vectors) can affect the results in the neutralizing assay. This argues that full spike alone should be used for these assays. I have no major concerns with this paper and only a few minor comments.
Page 3, line 98:
"As shown in Figure 1A, his system requires"
"his" should be "this"
Figure 3A. Is this an average of biological replicates (separate virus preps) or a single representative sample?
Page 6, line 178
"this assay requires only a relatively modest volume of virus is for a full 96-well plate neutralization assay"
delete "is"
Page 6, line 182
"As negative controls (not expected to neutralized)"
change "neutralized" to "neutralize"
4.3.4 It might be useful to specify the type of filter used. Group unfamiliar with lentiviral preps may accidentally use high-protein binding filters. And I assume you only tested a single freeze thaw. It might help to suggest that people aliquot stocks of the virus to prevent repeated freeze thaws.
4.4.6 Did you test whether polybrene is required? For MLV it is vital, but in our hands sometimes does not seem to affect lentiviral vectors. I agree it is best to use it unless it has been shown not to have an effect to remove it and I would not recommend holding up the publication to test this.
4.5.8 Perhaps there could be an additional line describing how you calculate IC50 using this python program, although it appears complete documention is available at the site.
Author Response
Comments and Suggestions for Authors
The Crawford et al manuscript is a description of a system for generating lentiviral particles pseudotyped by the SARS-Cov2 spike protein for use in neutralizing antibody assays. This is an important and timely article and will be very useful in the field. In addition to providing a very useful and detailed protocol along with easily accessible reagents, it shows that alterations of the tail of the spike protein (which has been previously done to increase titers of other coronavirus-pseudotyped virus vectors) can affect the results in the neutralizing assay. This argues that full spike alone should be used for these assays. I have no major concerns with this paper and only a few minor comments.
Page 3, line 98:
"As shown in Figure 1A, his system requires"
"his" should be "this"
Done.
Figure 3A. Is this an average of biological replicates (separate virus preps) or a single representative sample?
A single representative sample is shown. This is now noted in the figure legend.
Page 6, line 178
"this assay requires only a relatively modest volume of virus is for a full 96-well plate neutralization assay"
delete "is"
Done.
Page 6, line 182
"As negative controls (not expected to neutralized)"
change "neutralized" to "neutralize"
Done.
4.3.4 It might be useful to specify the type of filter used. Group unfamiliar with lentiviral preps may accidentally use high-protein binding filters. And I assume you only tested a single freeze thaw. It might help to suggest that people aliquot stocks of the virus to prevent repeated freeze thaws.
Thank you for this suggestion. We specified the filter type and added a note about aliquoting virus to avoid repeated freeze/thaw cycles.
4.4.6 Did you test whether polybrene is required? For MLV it is vital, but in our hands sometimes does not seem to affect lentiviral vectors. I agree it is best to use it unless it has been shown not to have an effect to remove it and I would not recommend holding up the publication to test this.
We have titered virus without polybrene and have noticed polybrene is not required for infection, but does increase titers slightly (<2-fold). We have not tested neutralization assays in the absence of polybrene.
4.5.8 Perhaps there could be an additional line describing how you calculate IC50 using this python program, although it appears complete documention is available at the site.
Neutcurve does have good documentation, but we did add more detail about how we processed the data prior to inputting it into neutcurve, and also explained the form of the curve (a Hill curve) fit by the program.
Reviewer 2 Report
The paper by Crawford et al., entitled “Protocol and reagents for pseudotyping lentiviral particles with SARS-CoV-2 Spike protein for neutralization assays” describes the technical procedure and the reagents for conducting serum neutralizing assays using lentiviral particles pseudotyped with the spike glycoprotein of SARS-CoV-2. With the ongoing Covid-19 pandemic caused by the emergence of SARS-CoV-2, strategies to combat the virus entry into targets human cells are essential to control the virus spread. For this, evaluation methods for measuring the inhibition of virus entry into target cells are essential. Serum neutralization assay is the pivotal technique for evaluating the proportion of neutralizing antibodies in the plasma of infected patient. However, the use of highly infectious and pathogenic virus to perform the assays is highly constringent. Pseudotyped lentiiviral with SARS-CoV-2 glycoproteins described in the present paper provides a safe and reliable way to perform the neutralizing assays. The authors have made all the reagents available in the BEI Resources repository of ATCC and the NIH.
Comment 1. Is there any reason why the authors choose plasmids without selectable markers to build the transcomplementing and indicator hACE2 293T-based cell line? Cell lines will lose expression of transfected genomes thereby will decrease or cancel production of spike-pseudotyped lentiviral particles and susceptibility to infection.
Comment 2. Competitive assays using ACE2 recombinant protein in the neutralization assay with the plasma of SARS-CoV-2 patient would have provided more insight on the specificity of the assay.
Comment 3. No comparison of the pseudotyped virions with the wild type virus has been described? Can the authors discuss the level of detection with the classical virus-based and this new pseudotyped particle-based neutralization assays.
Author Response
Comments and Suggestions for Authors
The paper by Crawford et al., entitled “Protocol and reagents for pseudotyping lentiviral particles with SARS-CoV-2 Spike protein for neutralization assays” describes the technical procedure and the reagents for conducting serum neutralizing assays using lentiviral particles pseudotyped with the spike glycoprotein of SARS-CoV-2. With the ongoing Covid-19 pandemic caused by the emergence of SARS-CoV-2, strategies to combat the virus entry into targets human cells are essential to control the virus spread. For this, evaluation methods for measuring the inhibition of virus entry into target cells are essential. Serum neutralization assay is the pivotal technique for evaluating the proportion of neutralizing antibodies in the plasma of infected patient. However, the use of highly infectious and pathogenic virus to perform the assays is highly constringent. Pseudotyped lentiiviral with SARS-CoV-2 glycoproteins described in the present paper provides a safe and reliable way to perform the neutralizing assays. The authors have made all the reagents available in the BEI Resources repository of ATCC and the NIH.
Comment 1. Is there any reason why the authors choose plasmids without selectable markers to build the transcomplementing and indicator hACE2 293T-based cell line? Cell lines will lose expression of transfected genomes thereby will decrease or cancel production of spike-pseudotyped lentiviral particles and susceptibility to infection.
We chose to transduce 293Ts with hACE2 via infection with VSV G-pseudotyped lentivirus packaging the hACE2 gene as this is generally a fairly stable way to express a gene of interest in target cells. We agree that transfected genomes can be readily lost by cells, but this method of transducing cells should be more stable than propagating transfected cells and allowed us to create 293T-ACE2 cells quickly. We did not include a fluorescent marker in our hACE2 plasmid to avoid potential conflicts with additional fluorophores in future experiments. Similarly, we did not use antibiotic selection because the gene is stably integrated into the genome by lentiviral transduction. Importantly, Fig. 2A confirms that expression of hACE2 in these cells remains high after many passages (up to ~12 passages and ~6 weeks post-sort at the time of writing). As such, we do not expect loss of hACE2 to be a big problem for the typical use of these cells. Nonetheless, we recognize that expression could decrease overtime and recommend that labs using these cells freeze down stocks of these cells at a low passage number and, if passaging for a long time, periodically confirm high levels of expression via staining for hACE2. We have added text to suggest this at ~line 300.
Comment 2. Competitive assays using ACE2 recombinant protein in the neutralization assay with the plasma of SARS-CoV-2 patient would have provided more insight on the specificity of the assay.
We think the reviewer is suggesting that the addition of ACE2 could confirm that the antibodies are actually targeting SARS-CoV-2 rather than some other part of the lentivirus. The issue is that soluble ACE2 also neutralizes the virus, so even if the ACE2 competes off the antibody the virus will still be neutralized (the reviewer’s suggested approach would work well with an ELISA assay). Therefore, we have chosen to validate the specificity by running pools of pre-2020 human sera and verifying that none of them neutralize the virus (Figure 4) like the convalescent human sera, suggesting that the neutralizing activity is specific to prior infection with SARS-CoV-2.
Comment 3. No comparison of the pseudotyped virions with the wild type virus has been described? Can the authors discuss the level of detection with the classical virus-based and this new pseudotyped particle-based neutralization assays.
We are in the process of directly comparing IC50s from our assay to IC50s obtained by another lab using a BSL-3 neutralization assay with the SARS-CoV-2 virus. We do not have those results yet (and do not think it is necessary to delay publication in order to get them), but the IC50s we have gotten with our pseudotyped-lentivirus neutralization assay are within the range of IC50 values that have typically been reported for other pseudotyped neutralization assays. In addition, in the Introduction we cite several references that suggest that pseudovirus titers generally correlated well with live virus titers for SARS-CoV-2, but also clearly acknowledge that live virus assays are the gold standard.
Reviewer 3 Report
Due to high infectivity and pathogenicity, SARS-CoV-2 should be handled in bio-safety level 3 (BSL-3) facilities, which has limited the accessibility of virus neutralization assay for developing candidate vaccines and therapeutics, especially neutralizing antibodies. Current studies reported that pesudovirus-based neutralization assay well correlates with measurement using live SARS-CoV-2 strain. This article established high-thought pesudovirus neutralization assay with pseudotype lentiviral particles with SARS-CoV-2 spike. Importantly, this study provided plasmids information, protocol and correlated reagents in detail, such that other groups could be available to perform the assay. Also, the distinct neutralization sensitivity was observed by modifying the cytoplasmic tail of SARS-CoV-2 Spike, suggesting that Spike protein without any cytoplasmic tail modifications should be used to perform the assay.
Major comments:
1. Most recombinant lentivirus production systems contain envelope plasmid, packaging plasmid and transfer plasmid, as described in the article. However, only two types of plasmids, including envelope plasmid and backbone plasmid was used in packaging of SARS-CoV-2 pseudotype lentiviral particles in some other researches (Emerg Microbes Infect. 2020 Dec;9(1):680-686. Cell Res. 2020 Apr;30(4):343-355. ). What the advantages of the method in this research than previously reported?
2. 1 ug of lentiviral backbone plasmids, 0.22 ug of packaging plasmids, and 0.34 ug of envelope plasmids were used to transfect 293T cells and viruses were harvested in supernatant. Did you optimized the ratios of these plasmids to get higher titers of viruses?
3. Generally, Huh-7 (human hepatoma cell) and Vero E6 (monkey kidney cell) cell lines were the natural target cells for SARS-CoV-2 and widely used in neutralization assay. If used the two cell lines as target cells, is it work for your measurement system?
4. As mentioned in Material and methods, the titer of pesudovirus could be calculated by flow cytometry or luciferase. Fluorescent signal of ZsGreen is detected by flow cytometry and relative luciferase units is detected using luciferase. It is confusing that the higher titers of relative luciferase units are observed using the Luciferase-IRES-ZsGreen backbone but the fluorescent signal of ZsGreen is lower.
Minor comments
1. Line 98, “his” should be this
2. line 156 and 158, Figure 1B should be “Figure 3B” and Figure 1C should be “Figure 3C .
3. Line 391, Fig.4 should be “Figure 4”.
4. Specify the “human IgG” in line 416.
5. Check reference 7, 38, 44
Author Response
Comments and Suggestions for Authors
Due to high infectivity and pathogenicity, SARS-CoV-2 should be handled in bio-safety level 3 (BSL-3) facilities, which has limited the accessibility of virus neutralization assay for developing candidate vaccines and therapeutics, especially neutralizing antibodies. Current studies reported that pesudovirus-based neutralization assay well correlates with measurement using live SARS-CoV-2 strain. This article established high-thought pesudovirus neutralization assay with pseudotype lentiviral particles with SARS-CoV-2 spike. Importantly, this study provided plasmids information, protocol and correlated reagents in detail, such that other groups could be available to perform the assay. Also, the distinct neutralization sensitivity was observed by modifying the cytoplasmic tail of SARS-CoV-2 Spike, suggesting that Spike protein without any cytoplasmic tail modifications should be used to perform the assay.
Major comments:
- Most recombinant lentivirus production systems contain envelope plasmid, packaging plasmid and transfer plasmid, as described in the article. However, only two types of plasmids, including envelope plasmid and backbone plasmid was used in packaging of SARS-CoV-2 pseudotype lentiviral particles in some other researches (Emerg Microbes Infect. 2020 Dec;9(1):680-686. Cell Res. 2020 Apr;30(4):343-355. ). What the advantages of the method in this research than previously reported?
The first paper referenced uses a VSV-pseudotyped system and the second paper uses an NL4-3 lentivirus system. All three systems (the two the reviewer mentions) and ours should work well. In fact, it is likely that the VSV system may allow for higher titers of SARS-CoV-2-pseudotyped virus than our system. We do have some anecdotal evidence that the lentiviral pseudotyping system we discuss here may work better than some other commonly used lentiviral systems, but we have not thoroughly tested all lentiviral systems, and many should work. The main advantage of our system is that we have made the protocol and reagents widely available to facilitate other researchers running this assay. Previous papers (included the two mentioned) have fairly sparse methods sections that would make it difficult for other researchers to adapt and use their assays. We explicitly acknowledge (third paragraph of introduction) that our pseudotyping system is not novel: rather the value is that it works well and can easily be implemented by others. It’s actually hard to compare to most other systems that have been described in the other references cited in this paragraph because most do not report details like the pseudovirus titers obtained.
- 1 ug of lentiviral backbone plasmids, 0.22 ug of packaging plasmids, and 0.34 ug of envelope plasmids were used to transfect 293T cells and viruses were harvested in supernatant. Did you optimized the ratios of these plasmids to get higher titers of viruses?
We slightly optimized these ratios so that we were adding the same molar amount of Spike plasmid as we had been adding of other entry protein plasmids for other lentiviral pseudotyping work going on in our lab. However, we did not test other ratios than these. These ratios work well and yield sufficient titers of lentivirus, but we acknowledge that further optimization of these ratios could potentially improve titers.
- Generally, Huh-7 (human hepatoma cell) and Vero E6 (monkey kidney cell) cell lines were the natural target cells for SARS-CoV-2 and widely used in neutralization assay. If used the two cell lines as target cells, is it work for your measurement system?
In very preliminary tests, we tried infecting Vero E6 cells with somewhat equivocal results (but things were not fully optimized), but got much better infection with the 293T-ACE2 cells, so moved forward with those. We have not tried infecting Huh-7 cells with this system, but have heard anecdotally that they also do not work as well as the 293T-ACE2 cells. In theory, any cell line susceptible to SARS-CoV-2 infection and that can be readily transduced by HIV-based lentiviruses should work for this system, but the 293T-ACE2 cells seem to work best.
- As mentioned in Material and methods, the titer of pesudovirus could be calculated by flow cytometry or luciferase. Fluorescent signal of ZsGreen is detected by flow cytometry and relative luciferase units is detected using luciferase. It is confusing that the higher titers of relative luciferase units are observed using the Luciferase-IRES-ZsGreen backbone but the fluorescent signal of ZsGreen is lower.
First, it is difficult to directly compare luciferase titers as measured in relative luciferase units per mL to ZsGreen titers as measured by transduction units (TU) per mL. One RLU per mL is not directly equivalent to one TU per mL. Indeed, a single cell transduced with the luciferase-containing backbone is likely contributing more than one RLU, whereas any cell transduced with a ZsGreen backbone can only be counted as one positive cell in the flow cytometry readout of ZsGreen. This fact explains why the luciferase RLUs are higher than the ZsGreen TU: a single virion can produce many RLUs. Furthermore, the ZsGreen signal is lower from the Luciferase-IRES-ZsGreen backbone than the ZsGreen only backbone because expression is typically decreased following an IRES compared to expression from a single-gene construct (such as the ZsGreen only backbone). Thus, the differences in ZsGreen expression between the backbones are a result of the differences in construct design rather than any biological differences in titer.
Minor comments
- Line 98, “his” should be this
Done
- line 156 and 158, Figure 1B should be “Figure 3B” and Figure 1C should be “Figure 3C .
Done
- Line 391, Fig.4 should be “Figure 4”.
Done
- Specify the “human IgG” in line 416.
We noted that it is the FC region of human IgG1
- Check reference 7, 38, 44
We have fixed these references.